# Preliminary Results on Preparation and Performance of a Self-Emulsifying Waterborne Epoxy Curing Agent at Room Temperature

**DOI:** 10.3390/polym15071673

**Published:** 2023-03-28

**Authors:** Zhenzhen Pi, Changyu Deng, Hongmei Pan, Kemei Pei

**Affiliations:** Department of Chemistry, Zhejiang Sci-Tech University, Hangzhou 310018, China

**Keywords:** epoxy resin, curing agent, polyethylene glycol, self-emulsifying

## Abstract

Polyethylene glycol 1000 (PEG1000) and epoxy resin E20 were used to synthesize the E20/PEG1000 polymer (EP1K), which was later transformed into a self-emulsifying water-based epoxy curing agent by reacting with m-Xylylenediamine (MXDA). The effects of molecular weight, the molar ratio of the raw materials, the catalyst dosage, and the different co-solvents on the properties of the prepared curing agent were systematically explored. The infrared absorption spectra of E20, EP1K, and the water-based epoxy curing agent were compared and analyzed. The coating properties of the waterborne epoxy varnish, which was based on water-based epoxy curing agents to emulsify and cure the resin E44, were systematically tested. The results demonstrated that with a molar ratio of 1:1:4 of PEG1000, E20, and MXDA, the boron trifluoride etherate (BF_3_·Et_2_O) as catalyst accounts for 0.3% of the total mass of E20 and PEG1000, and an applicable period of 3 h for the prepared varnish, the anti-corrosion performance, and mechanical properties of the coatings were excellent.

## 1. Introduction

Water-based coatings employed water instead of organic solvent as the dispersion medium, which had a low volatile organic compound (VOC) content [1]. Compared with traditional solvent-based coatings, waterborne coatings had numerous advantages, including low cost, easy to obtain, safety, and little environmental impact. In many water-based coatings, waterborne epoxy resin coatings have been widely concerned and applied due to their strong adhesion to heterogeneous substrates, as well as excellent corrosion resistance and mechanical properties [2,3]. However, the epoxy resin was insoluble in water and exhibited both hydrophobic and thermosetting properties [4,5]. The hydration of the epoxy resin and the application of the appropriate curing agent were the two issues that needed to be resolved if the epoxy resin was employed in a water-based epoxy coating in order to achieve the coating’s superior mechanical capabilities and corrosion resistance [6].

Water-based epoxy resin used water as the continuous phase, and the epoxy resin was dispersed in water by some method and mixed with water in the form of droplets or particles to form a stable and uniform emulsion [7]. Water-based epoxy resin and water-based epoxy curing agent were the two essential components of a water-based epoxy system. The ultimate performance of the cured film was determined by their structure and the effectiveness of the curing process. The reaction principle was that when the epoxy resin was cross-linked with various curing agents containing active hydrogen, a three-dimensional reticular structure was formed [8,9].

Curing agents also are known as hardeners, which are a class of substances that enhance or control curing reactions [10,11]. The theory behind solidified epoxy resin was that by adding a curing agent, the thermosetting resin goes through a sequence of chemical reactions that result in an irreversible alteration [12,13,14]. Water-based epoxy curing agent employed water as a diluent that contained less VOC, which thus had excellent environmental advantages [15,16]. Water-based epoxy curing agent was a significant component of the hydrous epoxy coating, and its structure and composition had a decisive role in the physical and chemical properties of the water-based epoxy coating [17]. The water-based epoxy curing agent with emulsifying and curing performance was one of the main varieties of water-based epoxy curing agents. It could not only emulsify and cure low-molecular-weight liquid epoxy resin, but also could be utilized with high-molecular-weight epoxy resin emulsion [18,19].

Polyethylene glycol (PEG) is non-toxic and environmentally friendly; having an excellent water solubility, its introduction into the epoxy resin molecular chain could result in water-based epoxy resin [20,21,22,23,24]. In this article, PEG1000 was utilized to construct self-emulsifying EP1K system, and then PEG1K was added to MXDA which converted poisonous MXDA into a non-toxic water-based epoxy curing agent. Moreover, the curing agent also had a good self-emulsifying function. The infrared absorption spectra of E20, EP1K, and the water-based curing agent were compared, and the coating properties of the waterborne epoxy varnish prepared based on the curing agent were systematically tested.

## 2. Experimental Section

### 2.1. Main Materials and Instruments

PEG (relative molecular weight: 800, 1000, 2000): analytical pure, Tianjin Chemio Chemical Reagent Co., Ltd. (Tianjin, China); epoxy resin (E20, E44): industrial grade, Zhejiang Tiannu Group Paint Co., Ltd. (Tongxiang, China); BF_3_·Et_2_O: analytical pure, Shanghai Aladdin Biochemical Technology Co., Ltd. (Shanghai, China); MXDA: analytical pure, Shanghai Maclin Biochemical Technology Co., Ltd. (Shanghai, China); propylene glycol methyl ether (PM): analytical pure, Shanghai Maclin Biochemical Technology Co., Ltd.(Shanghai, China); absolute ethanol: analytical pure, Sinopharm Chemical Reagent Co., Ltd. (Shanghai, China); mesityl oxide: analytical pure, Shanghai Maclin Biochemical Technology Co., Ltd. (Shanghai, China); benzyl alcohol: analytical pure, Shanghai Maclin Biochemical Technology Co., Ltd. (Shanghai, China); glycerol: analytical pure, Shanghai Macklin Biochemical Technology Co., Ltd. (Shanghai, China); ethylene glycol butyl ether: analytical pure, Shanghai Maclin Biochemical Technology Co., Ltd. (Shanghai, China); deionized water: homemade; perchloric acid (HClO_4_): analytical pure, Tianjin Damao Chemical Reagent Factory (Tianjin, China); glacial acetic acid (HAc): analytical pure, Tianjin Jiangtian Chemical Co., Ltd. (Tianjin, China); tetraethyl ammonium bromide (NEt_4_Br): analytical purity, Tianjin Jiangtian Chemical Co., Ltd. (Tianjin, China); trichlormethane (CHCl_3_): analytical purity, Tianjin Jiangtian Chemical Co., Ltd. (Tianjin, China); crystal violet: analytical pure, Shanghai Maclin Biochemical Technology Co., Ltd. (Shanghai, China)

Constant temperature heating agitator: DF-101S, Changzhou Putian Instrument Manufacturing Co., Ltd. (Changzhou, China); constant temperature drying oven: DHG-9076A, Shanghai Jinghong Experimental Equipment Co., Ltd. (Shanghai, China); Fourier transform infrared spectrometer: Nicolet 5700, Thermo Fisher Scientific, Waltham, MA, USA; adjustable film coated: KTQ-II., Dongguan Youtu Instrument Co., Ltd.(Dongguan China); portable pencil scratch hardness tester: QHQ-A Dongguan Jiuli Electronic Technology Co., Ltd. (Dongguan, China); T-type bending machine tester: WZJ-II., Tianjin Shibo Weiye Chemical Instrument Co., Ltd. (Tianjin, China); high-speed centrifuge: TG16-WS, Changsha Xiangzhi Instrument Co., Ltd. (Changsha, China); electrochemical workstation: CHI660E, Shanghai Chenhua Instrument Co., Ltd. (Shanghai, China); viscosity tester: NDJ-9S, Bangxi Instrument Technology Co., Ltd. (Shanghai, China).

### 2.2. Experimental Principle

The difference between E44 and E20, which are both epoxy resin of the bisphenol A type, and their difference lay in the different degrees of polymerization which resulted in various epoxy values. The basic parameters of E44 and E20 are displayed in Table 1. The molecular structure of bisphenol A epoxy resin is shown in Figure 1.

In this experiment, E20 and PEG1000 were used to realize the hydration of epoxy resin, which was followed by a reaction with MXDA’s active hydrogen to create a water-based epoxy curing agent. Ultimately, the E44 was cured with the self-made curing agent in order to determine the performance of it. The reaction mechanism of the prepared nonionic waterborne epoxy curing agent was as follows: two carbon atoms and one oxygen atom in the epoxy group were in the same plane to form a ternary ring structure. Due to the differences in the atoms or groups connected by two carbon atoms in the ternary ring, there was a distinction in the electronegativity between the two carbon atoms and the oxygen atoms, which led to the electrostatic polarization. This action increased the density of the electron clouds around the oxygen atoms, which made the oxygen atom vulnerable to the attack of electrophilic reagents, the carbon atom vulnerable to the attack of nucleophilic reagents, and finally broke the C-O bond. Finally, epoxy-based rings were opened. In Figure 2, the specific synthesis route is displayed.

### 2.3. Preparation of EP1K

In a nitrogen atmosphere, a four-mouth flask with a stirring paddle, a thermometer, a constant pressure drip funnel, and a spherical condenser tube was filled with PEG1000, E20, and a precise amount of solvent PM. The temperature was raised to 70 °C to dissolve it at 500 rpm and stir evenly. Afterwards, the BF_3_·Et_2_O was added at 70 °C. After 3 h of dripped, the 70 °C was kept for a while. Then, the temperature was raised to 95 °C and the reaction was maintained at that temperature for 9 h. Eventually, the temperature was decreased to 60 °C and the solvents were added and diluted to 50% of the solid content to obtain EP1K.

### 2.4. Preparation of Self-Emulsifying Waterborne Epoxy Resin Curing Agent

A certain proportion of MXDA was added to a four-mouth flask, which was equipped with a stirring paddle, a thermometer, a constant pressure drip funnel, and a spherical condenser tube, and was heated to 75 °C. Then, a certain amount of EP1K was added dropwise at 75 °C and kept it warm for a period of time to obtain the target product.

### 2.5. Preparation of Coatings

The tinplate sheets were used as coating substrates and sanded with 800-mesh sandpaper before being used, which was then washed with absolute ethanol and wiped clean. The self-made self-emulsifying non-ionic waterborne epoxy resin curing agent and E44 were combined at high speed according to the ratio of the epoxy group and active hydrogen equivalent of 0.9. Appropriate deionized water was added to prepare the water-based epoxy varnish. According to the standard “General preparation of paint film” (GB/T1727-1992), the coatings were prepared. The varnish was coated on the polished and cleaned tinplate sheet with a 50 μm coating rod and was placed in a constant temperature and humidity box at 25 °C for 7 days.

### 2.6. Characterization

The Nicolet 5700 Fourier-transformed infrared spectrometer was used to identify the IR distinctive absorption peaks of various compounds. A clear line-of-sight observation approach was used to assess the curing agents’ appearance status. The thermal stability, room-temperature stability, and low-temperature stability of the curing agent were tested by placing it in an environment of 60 °C, 25 °C, and −4 °C. To evaluate the centrifugal stability, a sample of curing agent diluted with water was placed in a centrifuge and spun at room temperature for 30 min at a speed of 3000 r/min. For the proper quantity of the water-based curing agent sample, it was mixed evenly with water to 20% of the solid content, and then placed for 24 h in order to test the dilution stability. The identical amount of curing agents and deionized water were both added into a beaker to ascertain the water solubility. The drying time was determined according to the standard “Methods of test for drying time of coatings, paints and putties” (GB/T1728-79). According to the standard “Paints and varnishes-Determination of resistance to liquids” (GB/T9274-88), acid, alkali, and water resistance were evaluated. Coating hardness was characterized according to the standard “Paints and varnishes determination of film hardness by pencil test” (GB/T6739-2006). Adhesion performance was tested according to the standard “Paints and varnishes-Cross cut test for films” (GB/T9286-1998). Impact resistance was tested according to the standard “Determination of impact resistance of the film” (GB/T1732-1993). The epoxy conversion rate was measured by the HClO_4_-HAc method. The CHI 660E electrochemical workstation was used to conduct the AC impedance (EIS) test. The sample’s viscosity was measured using an NDJ-9S viscosity tester. The value of porosity was estimated using the buoyancy method.

## 3. Results and Discussion

### 3.1. Infrared Spectroscopic Analysis

The Fourier infrared spectra of E20, EP1K, and the self-made waterborne epoxy curing agent are displayed in Figure 3. From the spectra of E20 and EP1K, it can be seen that the characteristic peak of the epoxy group at 912 cm^−1^ was weakened but did not disappear, indicating that, in the first stage, PEG and epoxy resin were partially ring-opened and the epoxy group was retained. A significant ether bond absorption peak at 1150 cm^−1^ shows that the alcohol hydroxyl group of PEG100 and the epoxy group successfully polymerized to produce more hydrophilic ether bonds. Because the initial reaction opened the ring of the epoxy group to generate hydroxyl, the intermolecular interaction strengthens with the increase in hydroxyl content, and a broad absorption peak of hydroxyl appeared at approximately 3400 cm^−1^. The epoxy characteristic peak of the self-made waterborne epoxy curing agent disappears, suggesting that the EP1K’s epoxy group underwent a ring-opening reaction in the reaction of the second step, which left no epoxy group residue. Additionally, despite the fact that the primary amine should have two peaks at 3400 cm^−1^ and 3500 cm^−1^, only one broad peak was formed here. The N-H stretching vibration absorption peak of the primary amine only showed up at 3520 cm^−1^ in the spectrogram of the water-based curing agent; this could be explained by the close proximity and the overlap of the absorption peaks. The N-H vibration absorption peak of secondary amines, which appeared at 3250 cm^−1^, indicates that MXDA and EP1K had successfully polymerized to form secondary amines. The study of the infrared spectrum revealed that the reaction process followed the anticipated reaction path.

### 3.2. Selection of Polyethylene Glycol Molecular Weight

By controlling a single variable, the molar ratio of PEG, E20, and MXDA with different molecular weights was 1:1:4. By conducting experimental studies employing 0.3% BF_3_·Et_2_O as a catalyst under constant reaction temperature and time, the optimal molecular weight of PEG was determined. The effects of various PEG molecular weights on the effectiveness of waterborne epoxy curing agents and coating films are displayed in Table 2.

Table 2 demonstrates that the stability of the curing agents and the water resistance of the coating films were significantly influenced by the molecular weight of PEG. When the molecular weight was too high, on the one hand, the amount of reactive functional groups such as a terminal hydroxyl group was low, the reaction activity was poor, the rate at which the epoxy resin opened its rings was low, the system contained more hydrophobic epoxy resin, and the product stability was subpar; on the other hand, the extended chain length caused different molecules to entangle and impede particle movement, which made the water-based curing agent less stable. The hydrophilicity of the product improved with the PEG molecular weight, while, at the same time, the produced film’s water resistance decreased because the intermolecular hydrogen bonding force was strengthened. Through comparative experiments, PEG1000 was selected as the hydrophilic chain segment to carry out the hydrolyzation of E20.

### 3.3. Effect of Molar Ratio of PEG1000 and E20 on Curing Agent Performance

The comprehensive properties of the curing agents were related to the proportion of hydrophilic and lipophilic chain segments. Figure 4 demonstrates the water solubility of curing agents with different hydrophilic segment contents under certain synthetic conditions. The aqueous solution contained micro-soluble matter when the molar ratio of PEG1000 to E20 was 0.5:1, which was due to the low contents of hydrophilic chain segments, low ring-opening rates of epoxy resin, and inadequate response. In addition, the curing agents were less hydrophilic when the system contained more hydrophobic epoxy resins. The curing agent aqueous solution was transparent when the molar ratio of PEG1000 to E20 was 1:1, indicating that the curing agents exhibited good water solubility at this molar ratio. However, the PEG content was too high, as too many hydrophilic groups in the curing agents can reduce its corrosion resistance and water resistance. Therefore, the molar ratio of PEG100 to E20 was chosen as 1:1 for the experiment.

### 3.4. Selection and Dosage of Catalyst

Although EP1K with a self-emulsifying property can be developed by employing epoxy groups and hydroxyl groups for reactions, this reaction was difficult to occur. BF_3_·Et_2_O is toxic, but its catalytic effect is good, and it is the most commonly used and easily obtained Lewis acid. Therefore, BF_3_·Et_2_O was selected as the catalyst in this paper and we avoided the adverse effects by controlling its dosage. Under the condition that the molar ratio of E20 to PEG1000 was 1:1, experiments were conducted by adding 0.1%, 0.3%, 0.5%, and 1% BF_3_·Et_2_O to the reaction system, respectively. The effect of the amount of BF_3_·Et_2_O on the epoxy conversion and stability of the curing agent was tested. The outcomes are displayed in Figure 5 and Table 3.

Figure 5 and Table 3 demonstrated that, when the amount of BF_3_·Et_2_O was too little and the dissociated H^+^ was insufficient, it resulted in low reaction rates for alcohols and epoxy resins and the reactions could not be carried out adequately. With the increase in BF_3_·Et_2_O dosages, the stability of the curing agent increased. When the BF_3_·Et_2_O dosages were approximately 0.3% of the total mass of PEG1000 and E20, the curing agents were stable for more than 30 days. When the dosage of the catalyst reached 0.5%, the reaction viscosity increased, and the stability of the curing agent was very poor. In addition, BF_3_·Et_2_O would also participate in the cross-linking reaction of epoxy. A lot of the dissolved H^+^ would cause the epoxy resin to undergo a self-polymerization reaction when there was too much BF_3_·Et_2_O, which formed the colloidal substance. All the experiments showed that the BF_3_·Et_2_O content was 0.3%, and the curing agent had good performance in all aspects.

### 3.5. Effect of E20 and MXDA Molar Ratio on Curing Agent Performance

There were many types of amine curing agents; most common were aromatic amines, fatty amines, and others. Aromatic amines’, due to the existence of the rigid benzene ring structure, cured film had better heat resistance, corrosion resistance, and mechanical strength than the cured film made with aliphatic amines [25,26]. Since most aromatic amines were solid, they needed to be heated to melting before being cured. Moreover, using aromatic amines alone had a high level of toxicity, and the products produced after curing were prone to turning yellow. As a result, aromatic amines could be modified to reduce their toxicity and optimize their performance. In this experiment, epoxy amine addition method was used to modify MXDA, which not only prepared a non-toxic water-based epoxy curing agent, but also improved its water solubility, extended the usage time of varnish, improved the reaction activity, and achieved the effect of room-temperature curing. The synthesis route is shown in Figure 6.

The molar ratio of E20 to MXDA primarily impacted the stability of the curing agents and the applicable period of the emulsion obtained by emulsifying E44. When the MXDA concentration was too low, the reaction between the primary amine and the epoxy group was insufficient, which left EP1K in the system, making it easier for the curing agent to layer. When the content of MXDA was too high, the application period of emulsion was too short due to the strong reaction activity of the system. Table 4 shows the curing films’ comprehensive properties of the curing agent and the cured film under the different MXDA contents. Figure 7 is a characterization diagram of acid resistance of cured films with different MXDA contents. It can be seen from the results that the stability of the curing agent and chemical resistance and mechanical properties of the cured film were the best when the molar ratio of PEG1000, E20, and MXDA was 1:1:4.

### 3.6. Effect of Different Co-Solvents on the Properties of the Curing Agent

During the preparation of the curing film, it was found that orange peel, fish eyes, shrinkage holes, shrinkage edges, and other phenomena would appear within the film’s surface. Several types of co-solvents were added to this experiment under the premise process optimization in order to lower viscosity, adjust surface tension, optimize coating surface state, and produce superior experimental results. After repeated tests, the coating leveling was the best in the coating surface state with the addition of 10% mesityl oxide. It could be uniform without bubbles, shrinkage, etc. The results of the experiment are displayed in Table 5 and Figure 8.

When no co-solvent was added, there were some small resin particles in the emulsion, resulting in an uneven coating surface and poor adhesion. Although the adhesion greatly increased with the addition of 5% mesityl oxide, the additive quantity was insufficient, and bubbles were likely to form. It can be clearly observed in Figure 8c that the coating surface was smooth and the adherence was good when the adding amount reached 10%. Too much solvent could lead to yellowing of the coating. Therefore, 10% mesityl oxide co-solvent of the curing agent mass was ultimately selected in this experiment.

### 3.7. Basic Parameters of the Self-Made Curing Agent

The basic parameters of the self-emulsifying non-ionic waterborne epoxy curing agent prepared in this experiment are shown in Table 6.

Table 6 and Figure 9 show that the self-made water-based epoxy curing agent had a solid content of 50 ± 5% and was a yellow transparent liquid. Its active hydrogen equivalent was 159 g/mol. At room temperature, the NDJ-9S viscosity tester yielded a reading of 4600 mpa·s for the viscosity of the curing substance. The curing agent had exceptional stability as evidenced by the fact that it remained stable for more than 180 days at room temperature.

### 3.8. Comparison between Self-Made Curing Agent and Commercially Available Curing Agent

In order to further systematically and intuitively understand the relevant properties of the self-emulsifying non-ionic waterborne epoxy curing agent prepared in this experiment, the coating properties of the self-made self-emulsifying non-ionic waterborne epoxy curing agent (CA1), a commercially available nonionic waterborne epoxy curing agent with E20 modified triethylenetetramine (CA2), and a commercially available ionic waterborne epoxy curing agent (CA3) were compared according to the molar ratio of active hydrogen to epoxy group 0.9:1. The coating surface and performance test results are shown in Figure 10 and Table 7.

Table 6 and Figure 10 show that the CA1- and E44-cured coatings performed much better overall than the CA2 and CA3. The cause might be linked to the fact that the CA1 with the addition of flexible hydrophilic segment PEG gave it good flexibility, allowing the coating to deform under rapid gravity without breaking or peeling. In addition, the porosity of the coating cures by the CA1 was significantly smaller than that of the coating cures by CA2 and CA3, which indicated that the CA1 coating showed excellent resistance when corrosive substances invaded the coating. Ionic waterborne epoxy curing agent CA3 was obtained by direct reaction of epoxy resin and amine curing agent. Due to the lack of flexible chain segments in the CA3, the surface brittleness of the coating was large, resulting in poor toughness and impact strength of the coating. In addition, the preparation of ionic waterborne epoxy curing agents requires the addition of organic acids to neutralize to produce soluble salts. The presence of free salts during the reaction process can lead to a decrease in the emulsification performance of this type of curing agent, and can affect the surface state and performance of the coatings.

### 3.9. Corrosion Resistance Test

Nyquist pictures of the coatings cured with two commercially available curing agents and the self-made curing agents in 3.5 wt% NaCl and 3.5 wt% KOH solutions were compared to further confirm the anti-corrosion effectiveness of the coatings. Figure 11 shows that the capacitive impedance of the varnish coating made by CA1 was much higher than that of the other two coatings, indicating that the self-made curing agent had better anti-corrosion performance.

## 4. Conclusions

In this paper, ring-opening polymerization reaction was carried out using E20 and PEG1000 as the raw ingredients. The self-emulsifying system EP1K was created by introducing a PEG1000 hydrophilic flexible chain segment. EP1K was then added to MXDA to create a non-toxic, room-temperature curable, non-ionic waterborne epoxy curing agent with self-emulsifying qualities. The experimental conditions for the preparation of EP and final products were also discussed. The IR spectra of E20, EP1K, and self-made waterborne epoxy curing agents were analyzed to verify the chemical reaction process. The factors affecting the stability of the curing agents, the mechanical properties, and corrosion resistance of the coating were studied through comparative experiments. Through the test and analysis of the comprehensive performance of the self-made waterborne epoxy curing agent and the emulsion curing E44 coating, the experimental formula and experimental conditions of the best performance were finally selected: the ratio of n (PEG1000):n (E20):n (MXDA) was 1:1:4, and the ring-opening polymerization was completed in two steps. Using PM as the solvent and 0.3 wt% BF_3_·Et_2_O as the catalyst, a waterborne epoxy curing agent was produced. Adding 10% mesityl oxide in the system could improve the leveling property of the coating. The curing agent had a solid content of 50 ± 5%, viscosity of 4600 mpa·s, active hydrogen equivalent of 159 g/mol, and storage stability of more than 180 days. The varnish was prepared by emulsifying and curing E44 with self-made water-based epoxy curing agent according to the ratio of active hydrogen/epoxy base 0.9:1. The hardness of the coating was 5 H, the impact strength was 50 kg·cm, the flexibility was 0 mm, the adhesion was grade 0, and the acid, alkali, and water resistance of the coating were qualified.

## Figures and Tables

**Figure 1 polymers-15-01673-f001:**
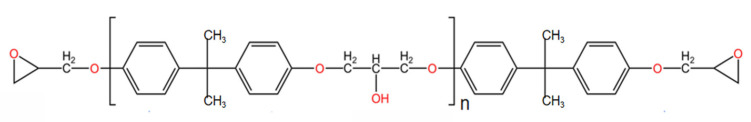
Molecular structure of bisphenol A epoxy resin.

**Figure 2 polymers-15-01673-f002:**
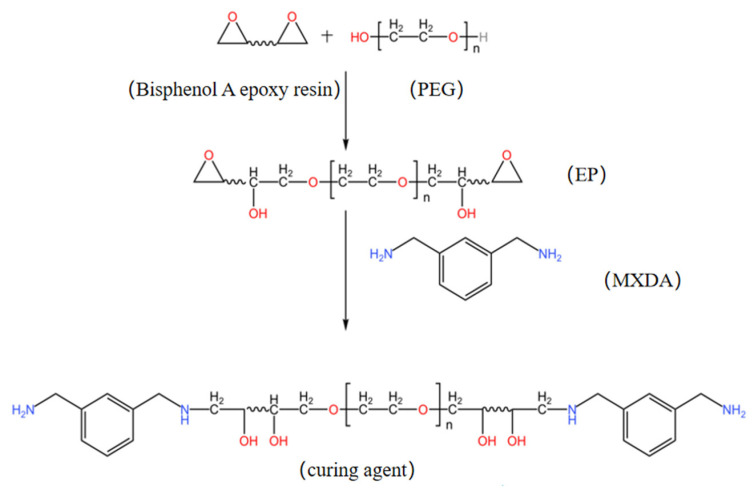
Synthesis route of a self-emulsifying non-ionic waterborne epoxy curing agent.

**Figure 3 polymers-15-01673-f003:**
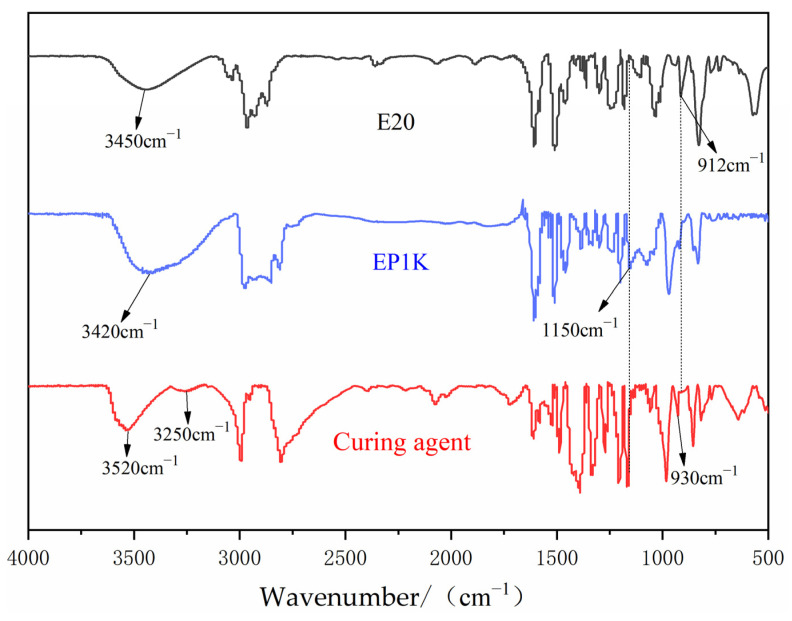
IR spectrum diagram of E20, EP, and self-made waterborne epoxy curing agent.

**Figure 4 polymers-15-01673-f004:**
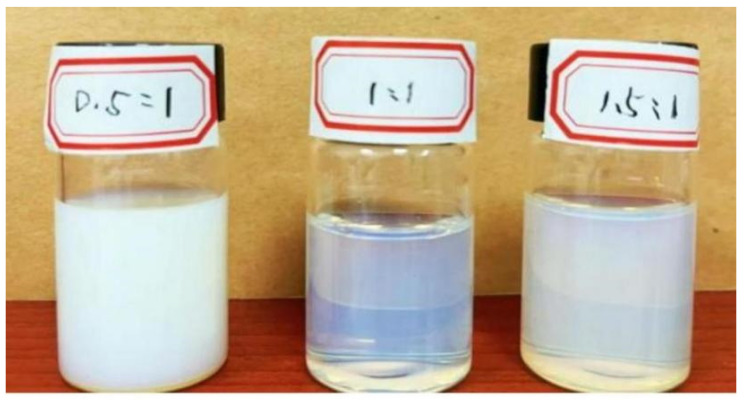
The effect of molar ratio of PEG1000 and E20 on the water solubility of curing agents.

**Figure 5 polymers-15-01673-f005:**
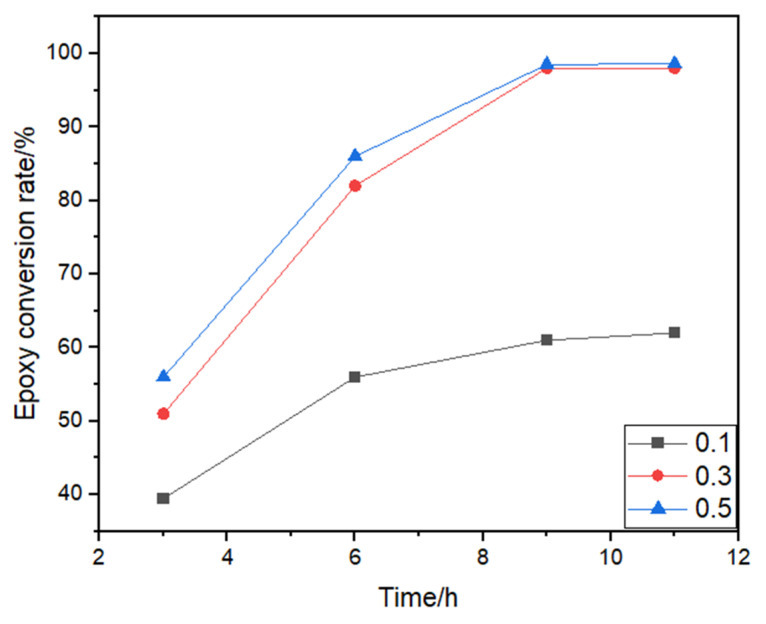
Effect of different amounts of BF_3_·Et_2_O on epoxy conversion rate. (Note: n (PEG1000):n (E20) = 1:1 when discussing the impact of catalyst dosage).

**Figure 6 polymers-15-01673-f006:**
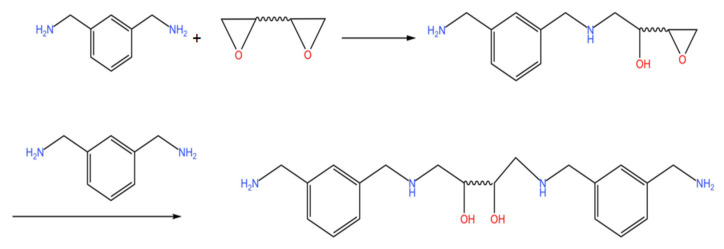
Synthesis route of a modified MXDA curing agent.

**Figure 7 polymers-15-01673-f007:**
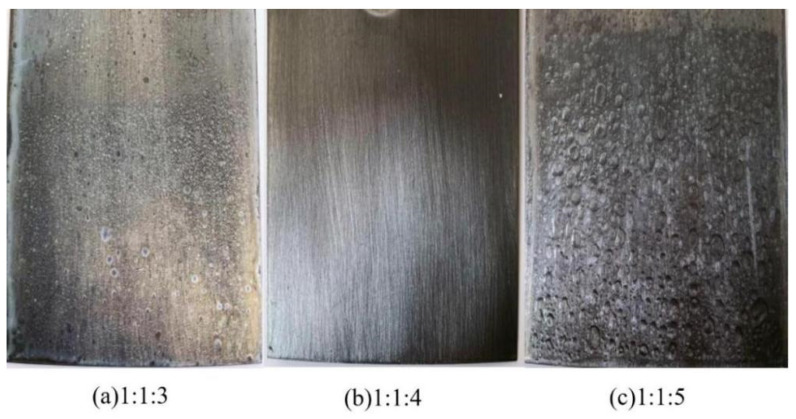
Characterization diagram of acid resistance of cured films with different MXDA content (24 h): (**a**): 1:1:3, (**b**): 1:1:4, (**c**): 1:1:5.

**Figure 8 polymers-15-01673-f008:**
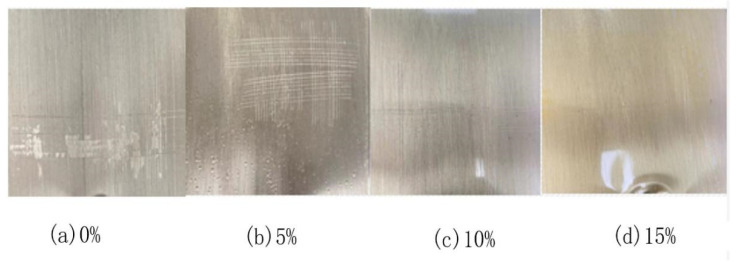
Coated surface with different contents of mesityl oxide: (**a**): 0%, (**b**): 5%, (**c**): 10%, (**d**): 15%.

**Figure 9 polymers-15-01673-f009:**
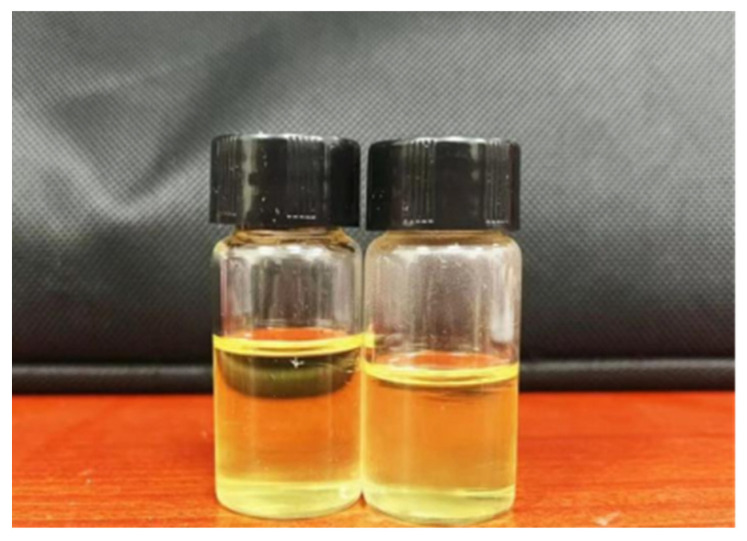
Self-emulsifying non-ionic waterborne epoxy curing agent.

**Figure 10 polymers-15-01673-f010:**
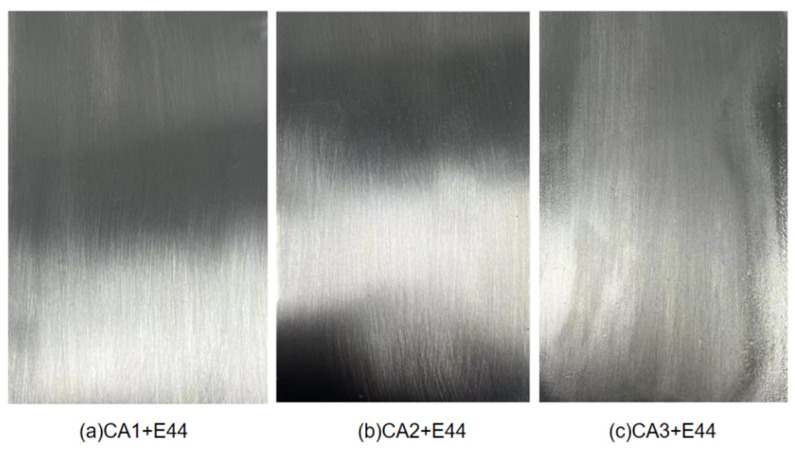
Coating surface of E44 cured with CA1 (**a**), CA2 (**b**), CA3 (**c**).

**Figure 11 polymers-15-01673-f011:**
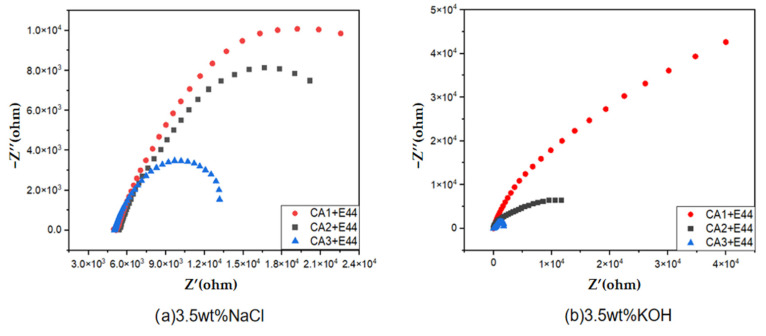
Nyquist curves of different coatings in 3.5 wt% NaCl (**a**) and 3.5 wt% KOH (**b**).

**Table 1 polymers-15-01673-t001:** Basic parameters of E44 and E20.

Epoxy Resin Name	E44	E20
Average epoxy value	0.44	0.20
Relative molecular mass	455	1000
Appearance	Colorless liquid	Light yellow solid

**Table 2 polymers-15-01673-t002:** Effect of PEG molecular weight on waterborne epoxy curing agent and film performance.

PEG Molecular Weights	800	1000	2000
Product appearance	Yellow liquid	Orange-yellow liquid	Yellow liquid
Thermal storage stability	30 days, slight layered	>30 days, stable	30 days, slight layered
Normal-temperature stability	3 days, layered	>30 days, stable	1 days, slight layered
Low-temperature stability	1 day, layered	>30 days, stable	1 day, layered
Centrifugal stability	10 min, layered	30 min, stable	20 min, layered
Dilution stability	24 h, slight layered	48 h, stable	24 h, stable
Water resistance	3 days, lightly whited	7 days, no changed	3 days, whited and blistered

**Table 3 polymers-15-01673-t003:** Effect of different amount of BF_3_·Et_2_O on the stability of the curing agents. (Note: n (PEG1000):n (E20) = 1:1 when discussing the impact of catalyst dosage).

BF_3_·Et_2_O Dosage (%)	0.1	0.3	0.5	1
Product appearance	Yellow liquid	Orange-yellow liquid	Yellow liquid	Colloidal substance
Viscosity (mpa·s)	3966	4600	12,223	-
Thermal storage stability	30 days, stable	>30 days, stable	30 days, stable	-
Low-temperature stability	24 h, light layered	48 h, stable	12 h, layered	-
Normal-temperature stability	3 days, layered	30 days, stable	2 days, light layered	-
Centrifugal stability	15 min, layered	30 min, stable	10 min, layered	-
Dilution stability	24 h, slight layered	48 h, stable	24 h, stable	-

**Table 4 polymers-15-01673-t004:** Effect of molar ratio of PEG1000, E20, and MXDA on the curing agent and the cured film performance.

Performance	n (PEG1000):n (E20):n (MXDA)		
	1:1:3	1:1:4	1:1:5
Emulsion trial period	<2 h	3 h	2–3 h
Coating surface drying time	5 h	4 h	4 h
Coating surface actual drying time	≤12 h	≤12 h	12–15 h
Acid resistance (5% HCl)	Not qualified	Qualified	Not qualified
Alkali resistance (5% NaOH)	Not qualified	Qualified	Not qualified
Flexibility (mm)	1	0	0
Impact resistant (50 kg·cm)	Not passed	Passed	Passed
Adhesion (Grade)	2–3	0	1–2
Pencil hardness	5 H	5 H	5–6 H

**Table 5 polymers-15-01673-t005:** Effects of the auxiliary solvent on the performance of the coating.

Co-Solvent	Benzyl Alcohol	Glycerol	Ethylene Glycol Butyl Ether	Mesityl Oxide
Coating appearance	Orange peel	Shrink edges	Shrink edges	Smooth
Porosity%	4.33	4.69	3.24	2.07

**Table 6 polymers-15-01673-t006:** Parameters of self-made self-emulsifying non-ionic waterborne epoxy curing agent.

Appearance	Solid Content/%	Active Hydrogen Equivalent/g·mol^−1^	Viscosity/mpa·s	Storage Stability at Room Temperature
Yellow transparent liquid	50 ± 5	159	4600	>180 days

**Table 7 polymers-15-01673-t007:** Coating properties of E44 cured with different curing agents.

Coating Type	CA1 + E44	CA2 + E44	CA3 + E44
Coating appearance	Smooth and even highlights	Smooth and even highlights	Rough and turbid surface
Pencil hardness	5 H	4–5 H	6 H
Flexibility (mm)	0	0	1
Impact strength(50 kg·cm)	Passed	Passed	Not passed
Porosity%	2.07	3.23	4.41
Adhesion (Grade)	0	1	1
Acid resistance (5% HCl)	Qualified	Qualified	Not qualified
Alkali resistance (5% NaOH)	Qualified	Qualified	Qualified
Water resistance/168h	Qualified	Qualified	Not qualified

## Data Availability

Not applicable.

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
