# Peer review of "Preliminary Results on Preparation and Performance of a Self-Emulsifying Waterborne Epoxy Curing Agent at Room Temperature"

_polymers, 2023, doi:10.3390/polym15071673_

Round 1
Reviewer 1 Report (Previous Reviewer 3)
1. Some statements in description of manuccript are not precise enough. For example:
- in line 140 - "The infrared structure of substances ...", What does it mean ?
- in line 158 - "Porosity measurement used Archimedes principle." What does it mean ?
- in line - "isopropyl fork acetone".
2. A quality of Tables 1 and 2 should be improved. Some lines should be separated in order to get clear informations.
3. In caption of Fig. 4 it should be added that it concerns the molar ratio of PEG1000 to E20.
4. In captions of Table 2 and Fig. 5 it should be added which PEG was used.
5. What is a difference between non-ionic waterborne epoxy curing agents CA1 and CA2 ? What it is "the ionic waterborne epoxy curing agent (CA3) ? What "the epoxy resin" and what "amine curing agent" were used for its preparation ? What were: the ratio of both reagents and reaction conditions ?
6. As authors are aware m-xylylenediamine (MXDA) is highly cancerogenic, so it seems that an application of MXDA to waterborne epoxy resins is not the best idea.
7. English language of this manuscript could be improved.
Author Response
- Some statements in description of manuscript are not precise enough. For example:
- in line 140 - "The infrared structure of substances ...", What does it mean?
lines 142-144:
"The infrared structure of substances ..."has been modified into “The Nicolet 5700 Fourier-transformed infrared spectrometer was used to identify the IR distinctive absorption peaks of various compounds’’.
- in line 158 - "Porosity measurement used Archimedes principle." What does it mean ?
lines 165-166:
"Porosity measurement used Archimedes principle" has been changed to “The value of porosity was estimated using the buoyancy method”
Using a scanning electron microscope, the apparent state of the coating was observed. Under the condition of 1 micron, no pores were observed in the coating. After extensive literature reading, it was found that the porosity was suitable for thermal sprayed coatings, glass fibers and other fields. However, the coating in this experiment was only a polymer film, and pores could not be observed. Even if there are pores in the coating we have made, it is not within the size range that can be observed by an electron microscope. Therefore, we used the buoyancy method to indirectly estimate the value of its porosity.
Buoyancy method:
The test steps are as follows:
(1) The coating shall be peeled off from the substrate, and there shall be no obvious unevenness in the appearance.
(2) Place the coating test piece in an oven at 30 ℃ for drying for about 2 hours to achieve a constant mass, and weigh the dry mass m1.
(3) Immerse the dried test piece in distilled water at room temperature and soak the exhaust gas under vacuum. After the bubbles are exhausted, take it out and weigh it. This test piece is called a saturated water test piece. Then, use a fine metal wire with a diameter of less than 0.5 mm to suspend the saturated water sample in water and weigh it. The weight value after deducting the mass of the metal wire is taken as the water mass m2 of the saturated water sample.
(4) Take the saturated water sample out of the water, quickly dry it with a wet cloth, and weigh it as the mass m3 of the saturated water sample. The surface porosity can be calculated by the following formula:
Surface porosity= (m3-m1)/(m3-m2) × 100%
- in line - "isopropyl fork acetone".
In full text," isopropyl fork acetone" has been modified into” Mesityl oxide”,
- A quality of Tables 1 and 2 should be improved. Some lines should be separated in order to get clear information.
Line 201and line 246:
Due to the addition of tables, Table 1 became Table 2 and Table 2 became Table 3, modifications have been made and the detailed results are shown in Table 2 and Table 3.
- In caption of Fig. 4 it should be added that it concerns the molar ratio of PEG1000 to E20.
line 235:
Figure 4. The effect of molar ratio of PEG1000 and E20 on the water solubility of curing agents.
- In captions of Table 2 and Fig. 5 it should be added which PEG was used.
lines 246-247and lines 264-265:
Table 2 becomes Table 3 due to the addition of tables, supplementary notes are added to the title in Table 3 and Figure 5(Note: n (PEG1000): n (E20) =1:1 when discussing the impact of catalyst dosage)
- What is a difference between non-ionic waterborne epoxy curing agents CA1 and CA2? What it is "the ionic waterborne epoxy curing agent (CA3)? What "the epoxy resin" and what "amine curing agent" were used for its preparation? What were: the ratio of both reagents and reaction conditions?
CA1 is a self-emulsifying non-ionic waterborne epoxy curing agent prepared by E20 modified MXDA in this experiment.
CA2 is a commercially available nonionic waterborne epoxy curing agent with E20 modified triethylenetetramine (TETA).
CA3 is a commercially available ionic waterborne epoxy curing agent that needs neutralization and salification.
CA2 and CA3 are commercially available curing agents are not products prepared by ourselves. Therefore, the specific synthesis conditions are not known. The performance and analysis of the coating cured by several curing agents are in the purpose is to compare the self-made water-based epoxy curing agent with other types of curing agents currently sold in the market, and the research value of this experimental product has been verified through experiments.
lines 343-349
In order to further systematically and intuitively understand the relevant properties of the self-emulsifying non-ionic waterborne epoxy curing agent prepared in this experiment, the coating properties of the self-made self-emulsifying non-ionic waterborne epoxy curing agent (CA1), a commercially available nonionic waterborne epoxy curing agent with E20 modified triethylenetetramine (CA2) and a commercially available ionic waterborne epoxy curing agent (CA3) were compared according for the molar ratio of active hydrogen to epoxy group 0.9:1.
- As authors are aware m-xylylenediamine (MXDA) is highly cancerogenic, so it seems that an application of MXDA to waterborne epoxy resins is not the best idea.
Traditional amine curing agents are toxic to varying degrees, and MXDA is toxic. It will cause great harm to human body if used alone as curing agent. However, modification with water-based epoxy resin can reduce the toxicity of MXDA, and the curing agent prepared in this experiment is non-toxic. In addition, because MXDA has a rigid structure of benzene ring, it can improve the mechanical properties of the coating. Through the addition of PEG, the curing agent prepared in this experiment also has a self-emulsifying function.
- English language of this manuscript could be improved.
The full text English has been modified and optimized.

Reviewer 2 Report (New Reviewer)
This work reports the “Preliminary results on preparation and performance of a self-emulsifying waterborne epoxy curing agent at room temperature.” In this manuscript, a few points must be included and/or clarified as major revisions before publication. The following comments are as follows:
1. Describe the abbreviations the first time. After that, the abbreviations should be used consistently throughout the whole manuscript.
2. The author should add the full form of VOC. The English language should be improved, and grammatical and typo errors should be fixed in the whole manuscript.
3. The sentence, “The epoxy resin itself was insoluble in water and had hydrophobicity and thermosetting properties.” should be cited by these references (https://doi.org/10.3390/polym14224828 and https://doi.org/10.3390/polym13071162). Lines 28-30 should be cited by some references.
4. Lines 34-35 and 42-45 should be rephrased.
5. The novelty of this work should be addressed in the last paragraph of the introduction.
6. The text of section 2.2 should be addressed in the results and discussion before section 3.1. The author should add more details about E44 and E20. Figure 2 should be revised again carefully. Is the marked red line correct for the R? How the O-O the bond form in EP?
7. Sections 2.3, 2.4, and 2.5 should be addressed in full detail. The complete form of EP1K should be addressed, although the authors have used different abbreviations (like EP) in Figure 2. In the FTIR description, the author states that the epoxy group had the characteristic peak near 910 cm-1, but the graph has no peak at 910 cm-1. The FTIR description should be improved; although the author described a fraction related to the FTIR data, the information is unclear.
8. Figure 5 shows the effect of different amounts of catalyst (0.1 to 0.5) on the epoxy conversion rate. Although, in table 2, the effect of the 1% catalyst has been addressed. The author should add the 1% catalyst data in figure 5. The author states, “When the dosage of the catalyst reaches 0.5%, the reaction viscosity increased, and the stability of the curing agents were very poor.” But there is no data regarding the viscosity in Table 2.
9. The sentence, “Aromatic polyamines provided good thermal stability due to the existence of rigid benzene ring structure. The heat resistance, corrosion resistance and mechanical strength of the curing film using this kind of curing agents were stronger than those of aliphatic amine curing film.” should be cited by the references (https://doi.org/10.1016/j.ijbiomac.2022.09.237 and https://doi.org/10.1021/acs.macromol.7b00097).
10. The discussion of section 3.7 should be increased. Lines 304-309 should be rephrased (the words, like in this chapter, should be rephrased).
11. The conclusion is too short, and some quantitative results should be included in the conclusion sections.
Author Response
1.Describe the abbreviations the first time. After that, the abbreviations should be used consistently throughout the whole manuscript.
This article has abbreviations for the first time in the professional terms, and abbreviations are used in the following text, has been modified in the full text
- The author should add the full form of VOC. The English language should be improved, and grammatical and typo errors should be fixed in the whole manuscript.
Line 24:
volatile organic compound (VOC).
Grammar and other issues have been revised in the full text.
- The sentence, “The epoxy resin itself was insoluble in water and had hydrophobicity and thermosetting properties.” should be cited by these references (https://doi.org/10.3390/polym14224828 and https://doi.org/10.3390/polym13071162). Lines 28-30 should be cited by some references.
Lines 29-30
However, epoxy resin was insoluble in water and exhibited both hydrophobic and thermosetting properties [4-5].
Lines 31-34
The hydration of the epoxy resin and the application of the appropriate curing agent were the two issues that needed to be resolved if the epoxy resin was employed in a water-based epoxy coating in order to achieve the coating's superior mechanical capabilities and corrosion resistance [6].
- Lines 34-35 and 42-45 should be rephrased.
Lines 35-37:
Water-based epoxy resin used water as the continuous phase, and the epoxy resin was dispersed in water by some method, and mixed with water in the form of droplets or particles to form a stable and uniform emulsion.
Lines 44-46
The theory behind solidified epoxy resin was that by adding a curing agent, the thermosetting resin goes through a sequence of chemical reactions that resulted in an irreversible alteration.
- The novelty of this work should be addressed in the last paragraph of the introduction.
Lines 57-63:
In this article, PEG1000 was utilized to construct self-emulsifying EP1K system, and then PEG1K was added to MXDA which converted poisonous MXDA into a non-toxic water-based epoxy curing agent. Moreover, the curing agent also had a good self-emulsifying function. The infrared absorption spectra of E20, EP1K and the water-based curing agent were compared, and the coating properties of waterborne epoxy varnish prepared based on curing agent were systematically tested.
- The text of section 2.2 should be addressed in the results and discussion before section 3.1. The author should add more details about E44 and E20. Figure 2 should be revised again carefully. Is the marked red line correct for the R? How the O-O the bond form in EP?
Lines 96-97:
The basic information of E20 and E44 is supplemented in Table 1 on page 3.
Page 3(Figure 2):
The O-O in Figure 2 is wrong and has been corrected. Due to the long intermediate chain of epoxy resin, I use wavy lines instead. The specific structure of epoxy resin is illustrated in Figure 1, so it is expressed in simplified form in Figure 2.See Figure 2 for specific results
- Sections 2.3, 2.4, and 2.5 should be addressed in full detail. The complete form of EP1K should be addressed, although the authors have used different abbreviations (like EP) in Figure 2. In the FTIR description, the author states that the epoxy group had the characteristic peak near 910 cm-1, but the graph has no peak at 910 cm-1. The FTIR description should be improved; although the author described a fraction related to the FTIR data, the information is unclear.
Sections 2.3, 2.4, and 2.5:
In Sections 2.3, 2.4, and 2.5, EP has been changed to EP1K.The complete form of EP1K is E20/PEG1000 polymer. The abbreviation appears for the first time in the abstract, so in the following text, the abbreviation (EP1K) was used.
Lines 169-189:
The infrared spectrum is retested as shown in Figure 3 and re-described
- Figure 5 shows the effect of different amounts of catalyst (0.1 to 0.5) on the epoxy conversion rate. Although, in table 2, the effect of the 1% catalyst has been addressed. The author should add the 1% catalyst data in figure 5. The author states, “When the dosage of the catalyst reaches 0.5%, the reaction viscosity increased, and the stability of the curing agents were very poor.” But there is no data regarding the viscosity in Table 2.
Lines246-262
Due to the addition of a table, Table 2 has become Table 3. In addition, the viscosity test data has been supplemented in Table 3.
In Table 3 and Figure 5, since the system obtained by adding 1% catalyst in the experiment will produce gel, which belongs to the non-liquid form, its data cannot be measured, this status cannot be applied and has no use value and we think such data may be meaningless. we hope that the reviewer will consider it.
- The sentence, “Aromatic polyamines provided good thermal stability due to the existence of rigid benzene ring structure. The heat resistance, corrosion resistance and mechanical strength of the curing film using this kind of curing agents were stronger than those of aliphatic amine curing film.” should be cited by the references (https://doi.org/10.1016/j.ijbiomac.2022.09.237 and https://doi.org/10.1021/acs.macromol.7b00097).
Lines 282: Relevant literature has been cited.
- The discussion of section 3.7 should be increased. Lines 304-309 should be rephrased (the words, like in this chapter, should be rephrased).
Line 332:”in this chapter” has been changed to “in this experiment “
Line 336-340: It is explained and supplemented below Table 6 and Figure 3.7
- The conclusion is too short, and some quantitative results should be included in the conclusion sections.
Line 378-399.
In this paper, ring-opening polymerization reaction was carried out using E20 and PEG1000 in a 1:1 molar ratio as the raw ingredients. The self-emulsifying system EP1K was created by introducing a PEG hydrophilic flexible chain segment. EP1K was then added to MXDA to create a non-toxic, room-temperature curable, non-ionic waterborne epoxy curing agent with self-emulsifying qualities. The experimental conditions for the preparation of EP and final products were also discussed. The IR spectra of E20, EP1K and self-made waterborne epoxy curing agents were analyzed to verify the chemical reaction process. The factors affecting the stability of the curing agents, the mechanical properties and corrosion resistance of the coating were studied through comparative experiments. Through the test and analysis of the comprehensive performance of the self-made waterborne epoxy curing agent and the emulsion curing E44 coating, the experimental formula and experimental conditions of the best performance were finally selected: the ratio of n (PEG1000): n (E20): n (MXDA) was 1:1:4 , and the ring-opening polymerization was completed in two steps. Using PM as solvent and 0.3 wt% BF3·Et2O as the catalyst, a waterborne epoxy curing agent was produced. Adding 10% mesityl oxide in the system could improve the leveling property of the coating. The curing agent has a solid content of 50 ± 5%, viscosity of 4600 mpa·s, active hydrogen equivalent of 159 g/mol, and storage stability of more than 180 days. The varnish was prepared by emulsifying and curing E44 with self-made water-based epoxy curing agent according to the ratio of active hydrogen/epoxy base 0.9:1. The hardness of the coating was 5H, the impact strength was 50kg·cm, the flexibility was 0 mm, the adhesion was grade 0, and the acid, alkali and water resistance of the coating were qualified.
In addition, each sentence of the full text has been optimized. Due to the large number of modifications to the full text syntax, it is convenient to view the revised content replied by the reviewer, and there is no tracking mark for the syntax modifications of the full text. We appreciate the efforts of the editors and reviewers.

Round 2
Reviewer 2 Report (New Reviewer)
Accept.
This manuscript is a resubmission of an earlier submission. The following is a list of the peer review reports and author responses from that submission.
Round 1
Reviewer 1 Report
The MS entitled „Preparation and performance of a self-emulsifying waterborne epoxy curing agent at room temperature“ written by Zhenzhen Pi, Changyu Deng, and Kemei Pei focusses on a water born epoxy curing agent, which was synthesized by reaction of a polyethylene glycol epoxy resin prepolymer E20 with m-Xylylenediamine. Analysis of the product was carried out by FT-IR spectroscopy. The product was applied to emulsify the resin E-44 for application as varnish. After coating the varnish on a substrate and curing, selected application tests were carried out, and the results were discussed.
Unfortunately, no chemical structures and no chemical equations are given in the MS. Therefore, understanding of the MS is difficult for the reader.
Furthermore, the MS is very descriptive. Information regarding the chemical structure of the synthesized product is poor.
The use of further analytical methods to characterize the synthesized product, to follow the curing process and to investigate the cured material may help to improve the scientific discussion of results.
I cannot recommend the MS for publication in MDPI polymers in the present form.
Reviewer 2 Report
There is no novelty in the work presented here.The detailedcharacterization for the waterborne systems needs to be explored such as storage stabilty, particle size/distribution etc. Detailed anticorrosion studies and plots such as nquist or Bode`s plot must be included as it is being claimed as anticorrosion coatings.
Reviewer 3 Report
1. I suggest to provide in Introduction more informations concerning waterborne epoxy systems.
2. What are chemical compositions of epoxy resins E22 and E44 ?
3. Which kind of epoxy resin was used for preparation of a propolymer EP ? E20 ? What reactions occured between the epoxy resin and PEG1000 ? What was a chemical structure of the propolymer EP ?
4. Analyses of IR spectra do not provide convincing informations for this kind of research project.
5. m-Xylylenediamine (MXDA) is a cancerogenic reagent, and a BF3-ether complex is also an agresive chemical.
6. Was porosity of epoxy coatings analyzed somehow ?
7. I agree with authors that a quality of prrepared waterborne epoxy coatings was poor.
8. The kind of co-solvent cannot affect the properties of curing agent (1), but itmust affect a quality of the epoxy coatings: leveling, porosity, etc.
9. It seems quite obvious that without addition of isopropyl acetone it was impossible to obtain the good quality epoxy coatings. So, it seems that other kinds of the epoxy resins (solventless and low viscosity systems) have better perspectives for practical applications, e.g. as anticorrosion coatings.
10. Finally, based on the quality of experimental results, I suggest to change a title for the following: "Preliminary results on preparation and performance of a self-emulsifying waterborne epoxy curing agent at room temperature"
11. English language and a style of manuscript description should be improved significantly, because some statements are not clear enough, for example: "alcohol ether solvents", "polyethylene glycol with average molecular weight 1000 ring-opening reaction", "and the ring opening and expansion rate with epoxy resin is low", "epoxy ring opening with epoxy resin E20".
I also suggest to use more often a Past Tense, instead of Present Tense.